# Molecular Investigation of Small Ruminant Abortions Using a 10-Plex HRM-qPCR Technique: A Novel Approach in Routine Diagnostics

**DOI:** 10.3390/microorganisms12081675

**Published:** 2024-08-14

**Authors:** Ioannis Gouvias, Marios Lysitsas, Apostolos Batsidis, Sonia Malefaki, Dimitra Bitchava, Anna Tsara, Emilija Nickovic, Ilias Bouzalas, Eleni Malissiova, Raphaël Guatteo, George Valiakos

**Affiliations:** 1Faculty of Veterinary Science, University of Thessaly, 43100 Karditsa, Greece; igouvias@uth.gr (I.G.); mlysitsas@uth.gr (M.L.); emilija.nickovic@vet.bg.ac.rs (E.N.); 2Department of Mathematics, University of Ioannina, 45110 Ioannina, Greece; abatsidis@uoi.gr; 3Department of Mechanical Engineering and Aeronautics, University of Patras, 26500 Rion-Patras, Greece; smalefaki@upatras.gr; 4Vet in Progress Plus, Veterinary Laboratories, 15343 Athens, Greece; bitchava.dimitra@vetinprogress.gr (D.B.); lab1@vetinprogress.gr (A.T.); 5Faculty of Veterinary Medicine, University of Belgrade, Bul. Oslobodjenja 18, 11000 Belgrade, Serbia; 6Veterinary Research Institute, Hellenic Agricultural Organization DIMITRA (ELGO-DIMITRA), Campus Thermi, 57001 Thessaloniki, Greece; bouzalas@elgo.gr; 7Food of Animal Origin Laboratory, Animal Science Department, University of Thessaly, 41500 Larissa, Greece; malissiova@uth.gr; 8BIOEPAR, INRAE Oniris, 44300 Nantes, France; raphael.guatteo@oniris-nantes.fr

**Keywords:** abortogenic pathogens, *Anaplasma phagocytophilum*, *Brucella* spp., *Campylobacter fetus*, *Chlamydophila* spp., *Coxiella burnetii*, Greece, HRM, *Neospora caninum*, qPCR, *Salmonella* spp., small ruminants, *Toxoplasma gondii*

## Abstract

The objective of this study was to apply and preliminarily evaluate a High-Resolution Melting (HRM) analysis technique coupled with qPCR, that allows the simultaneous detection of 10 different ruminant abortogenic pathogens, for investigating abortions in sheep and goats throughout Greece. A total of 264 ovine and caprine vaginal swabs were obtained the week following the abortion from aborted females and analyzed using a commercially available kit (ID Gene™ Ruminant Abortion Multiplex HRM, Innovative Diagnostics). Results indicated a high prevalence of *Coxiella burnetii* and *Chlamydophila* spp., which were detected in 48.9% and 42.4% of the vaginal swabs, respectively. Results for these most commonly detected pathogens were compared with those of a well-established commercial qPCR kit, with near-perfect agreement. *Toxoplasma gondii*, *Salmonella* spp., *Brucella* spp., *Anaplasma phagocytophilum*, *Campylobacter fetus,* and *Neospora caninum* were also identified, the two latter reported for the first time in the country in small ruminants. Mixed infections occurred in 35.6% of the animals examined. This technique allows for the simultaneous detection of many abortogenic pathogens in an accurate and cost-effective assay. Detection of uncommon or not previously reported pathogens in various cases indicates that their role in ovine and caprine abortions may be underestimated.

## 1. Introduction

Dealing with abortions in sheep and goats is extremely challenging. The co-implication of several agents in relevant cases and the diversity in their diagnosis and treatment interferes with their effective management at the farm level. Moreover, failure to successfully restrain an outbreak results in significant economic loss and animal welfare issues [1,2]. There are various possible causes of abortions [1,3]; however, the most crucial are infectious abortogenic agents because of their great impact on the health status of the flock and the danger they pose to public health, many of these agents being zoonotic [4].

A common cause of abortions and reproductive disorders in sheep and goats is *Coxiella burnetii*, an obligatory intracellular bacterium that is distributed worldwide on farms of domestic ruminants [5,6]. Goats seem to be more susceptible to *C. burnetii* than sheep [7]. Moreover, it is a significant zoonotic agent that causes Q fever in human hosts. Q fever outbreaks and a recent large-scale epidemic in the Netherlands were associated with close proximity to aborting small ruminants, which are considered reservoir hosts and the main source of *C. burnetii* human infection [8,9]. In addition, under optimal climatic conditions, transmission can be widespread, since airborne dispersal kernels have been estimated to be even 10 km around sheep and goat farms [10].

*Chlamydophila* spp., especially *C. abortus*, is also one of the main causes of infectious abortions in sheep and goats worldwide, and is the etiologic agent of enzootic abortion in ewes (EAE) or ovine enzootic abortion (OEA) [11]. Moreover, it has zoonotic potential, since infection can develop into a severe, life-threatening disease in pregnant women and can lead to severe respiratory disorders [12].

Several other pathogens have also been associated with abortion or stillbirth in small ruminants. *Brucella* spp., *Campylobacter fetus*, *Listeria monocytogenes*, *Leptospira* spp., *Salmonella* spp., *Toxoplasma gondii*, *Anaplasma phagocytophilum*, *Neospora caninum*, specific viruses (Border Disease virus, Rift Valley Fever virus, etc.), and fungi (*Aspergillus* spp.) have been identified, more or less frequently, in relevant cases [1,13,14,15].

Investigation of infectious abortions in sheep and goats includes clinical findings, microbiological cultures, serological tests, molecular assays, and histopathological examination of aborted tissues [1]. The co-implication of various infectious and non-infectious factors, the cost, the duration of the required tests as well as the fear of regulatory consequences in case of detection of zoonotic agents are limiting factors, inhibiting accurate etiological diagnosis in a significant percentage of cases [1,16]. Therefore, dealing with ovine and caprine abortions is frequently a “gordian knot” for veterinarians; the establishment of multiplex, accurate, fast, and low-cost differential diagnostic tools is vital.

High-resolution melting (HRM) analysis is a single-step closed-tube method that utilizes qPCR equipment to investigate DNA sequence variations [17]. It is a rapid, low-cost technique with excellent discriminatory power between molecular targets that demonstrate smaller or greater diversity [18,19]. It has been utilized in pathogen genotyping, monitoring the dynamics of microbial populations, managing outbreaks, and performing epidemiological surveillance [18]. In veterinary medicine, several possible applications of relevant techniques have been proposed in the literature [20,21,22]. The potential of HRM analysis has been evaluated for the simultaneous detection of abortogenic pathogens in ruminants, with promising results [23,24].

The objective of this study was to apply and preliminary evaluate a novel HRM analysis technique coupled with qPCR as an accurate and rapid diagnostic tool for the simultaneous detection of various abortogenic pathogens in sheep and goats.

## 2. Materials and Methods

### 2.1. Sample Collection

Vaginal swab samples were obtained by veterinary practitioners between May 2023 and May 2024 from ovine and caprine farms with reported abortive cases. These farms are distributed throughout mainland Greece as well as on islands and apply an intensive or semi-intensive system. Most of the farms have open-front buildings with earthen floors and they have small paddocks next to the main farm for exercise and grazing. Samples were obtained during the week following the abortion from the relevant animals, according to EFSA recommendations [25]. All swabs were maintained at 2–5 °C for no more than 48 h prior to further processing. All the owners of the farms visited consented to participate in the study. The animal study and informed consent protocol has been approved by the Animal Use Ethics Committee of the Faculty of Veterinary Medicine, University of Thessaly (Approval No. 176/18-3-2024).

### 2.2. Sample Preparation and Nucleic Acid Extraction

Each swab was coded and aseptically added to 15 mL conical centrifuge tubes containing 1 mL of sterile phosphate buffer saline (PBS). Subsequently, each tube was vortexed three times for 10 s and a quantity of 50 μL was obtained in duplicate for the extraction phase.

Whole DNA extraction was performed using an IDEAL™ 32 extraction robot (Innovative Diagnostics, Grabels, France) and a compatible commercial kit (ID Gene^TM^ Mag Fast Extraction Kit, Innovative Diagnostics, Grabels, France) and, in a few cases, a commercial spin column kit (IndiSpin Pathogen Kit, INDICAL BIOSCIENCE GmbH, Leipzig, Germany). All procedures were performed according to the manufacturer’s instructions.

The concentration of DNA in all obtained extracts was measured using a fluorometer (Invitrogen Qubit^TM^ 4, Thermo Fisher Scientific, Waltham, MA, USA), and only DNA extracts with measurable yields were further processed.

### 2.3. Molecular Assays

All samples were investigated for the presence of ten different abortifacient pathogens, which constitute main bacterial and parasitic infectious agents in domestic small ruminants.

Initially, the samples were tested using novel multiplex HRM analysis coupled with qPCR. The results for *C. burnetii* and *Chlamydophila* spp. were confirmed for all DNA extracts (positive or negative) with a well-established commercial qPCR assay, whereas DNA extracts that were positive for other pathogens (except *C. burnetii* and *Chlamydophila* spp.) were sent for confirmation in a private diagnostic laboratory, where qPCR was performed with commercially available well-established kits. All procedures carried out in this study are described in detail below.

#### 2.3.1. Multiplex HRM-qPCR Analysis

A novel commercial kit (ID Gene™ Ruminant Abortion Multiplex HRM, Innovative Diagnostics, Grabels, France) was used according to the manufacturer’s instructions on an MIC qPCR cycler (Bio Molecular Systems, Upper Coomera, Australia). For each reaction, 5 μL of sample extract and controls were added in two wells containing 10 μL of two different amplification reaction mixes (ARMs), A and B. ARM A is used for simultaneous qualitative detection of *A. phagocytophilum*, *Chlamydophila* spp., *C. burnetii*, *T. gondii*, *Salmonella* spp., and real-time monitoring of *C. burnetii* amplification. ARM B allows simultaneous qualitative detection of *L. monocytogenes*, *N. caninum*, *Leptospira* spp., *C. fetus*, *Brucella* spp., and real-time monitoring of *Brucella* spp. amplification. The PCR conditions were as follows: 95 °C for 2 min, followed by 40 cycles of 95 °C for 10 s and 64 °C for 30 s. HRM analyses were performed from 65 °C to 97 °C at a temperature gradient of 0.25 °C/s. The results were analyzed using the HRM data interpretation software DISoft^TM^ version 4.5.0 (Innovative Diagnostics, Grabels, France).

#### 2.3.2. qPCR Assays

A qPCR assay for the qualitative detection of *C. burnetii* and *Chlamydophila* spp. was performed using a commercial kit (ID Gene™ Q Fever-*Chlamydophila* spp. Triplex, Innovative Diagnostics, Grabels, France), according to the manufacturer’s guidelines on an MIC qPCR cycler (Bio Molecular Systems, Upper Coomera, Australia). Briefly, the conditions were as follows: 95 °C for 2 min, followed by 40 cycles at 95 °C for 10 s and 60 °C for 30 s, and the fluorescence signal was read at the end of the elongation phase. The results were interpreted according to the criteria provided in the kit’s manual. The threshold for differentiating between positive and negative results was set at Ct = 40. In order to have a preliminary verification of the HRM-qPCR assay detection results, and considering study cost limitations, DNA extracts of HRM-qPCR samples positive to pathogens other than *C. burnetii* and *Chlamydophila* spp. were sent to a private diagnostic laboratory for verification. The samples were analyzed using commercially available qPCR Kits for detection of *T. gondii*, *A. phagocytophilum*, *C. fetus*, *Neospora* spp. (Genetic Analysis Strategies SL, Orihuela, Spain), *Salmonella* spp., and *Brucella* spp. (BactoReal^®^, Ingenetix GmbH, Wien, Austria) according to the manufacturer’s instructions on a LightCycler 2.0/Roche Real-Time System (Roche Diagnostics International AG, Rotkreuz, Switzerland).

### 2.4. Statistical Analysis

Descriptive statistics were used to summarize the results obtained. Fischer’s exact test was performed to compare the results among animal species. Regarding the two most prevalent pathogens detected (*C. burnetii* and *Chlamydophila* spp.), the following methodology was utilized for evaluating the HRM-qPCR technique. Even though a gold-standard technique was not used, the HRM-qPCR results for *C. burnetii* and *Chlamydophila* spp. were compared with results obtained from a well-established commercially available qPCR kit (ID Gene™ Q Fever-*Chlamydophila* spp. Triplex, Innovative Diagnostics, Grabels, France). Initially, the agreement or concordance between the measurements of the two different techniques (HRM-qPCR and qPCR Triplex) was studied. In this context, the values of different indices of agreement, such as the proportion of observed agreement, positive (negative) agreement, average positive (negative) agreement, and Cohen’s kappa were computed [26,27,28], and McNemar’s test was performed [29]. McNemar’s test was utilized to detect statistically significant discrepancies between the numbers of positive and negative results for each paired dichotomous sample obtained by the two techniques. If the null hypothesis is not rejected, there is insufficient evidence to confirm the difference between the two techniques. Subsequently, the problem of testing equivalence or non-inferiority in a matched-pairs design with binary responses and without access to a gold standard was considered. According to Liu et al. [30], the assessment of equivalence can generally be divided into two major objectives: an equivalence test and a non-inferiority test. The problem of testing equivalence or non-inferiority in a matched-pairs design with binary responses is a common issue in medical research (e.g., Jin et al. [31] and the references therein). In this context, p_1_ and p_2_ denote the probabilities of positive results based on HRM-qPCR and qPCR Triplex, respectively, while δ is a predetermined clinically meaningful equivalence limit. Even though a common value for δ is 0.1, we used the value of 0.05 in order for the margin of differentiation between the two techniques to be smaller. Then, the noninferiority and equivalence tests can be stated in terms of the difference Δ = p_1_ − p_2_ [30,32]. According to Liu et al. [30], the equivalence between alternative and reference procedures can be tested in terms of the following interval hypotheses:H_0_: p_1_ − p_2_ ≥ δ or p_1_ − p_2_ ≤ −δ versus H_1_: −δ < p_1_ −p_2_ < δ.

The interval hypotheses are composed into two sets of one-sided hypotheses
H_01_: p_1_ − p_2_ ≤ −δ versus H_11_: p_1_ − p_2_ > −δ and H_02_: p_1_ − p_2_ ≥ δ versus H_12_: p_1_ − p_2_ < δ.

Instead of testing the previous hypotheses, the 100 (1 − 2a)% confidence interval of p_1_ *−* p_2_ can be computed. If the confidence interval of the difference lies within (−δ, δ), then the two tests can be considered equivalent at significance level a, where in our study we set a = 0.05.

IBM SPSS Statistics for Windows, version 29.0 (IBM Corp., New York, NY, USA), and NCSS Software, free trial version accessed on 10 July 2024, https://www.ncss.com/download/ncss/free-trial/, (NCSS LLC, Kaysville, UT, USA), were used for the statistical analysis.

## 3. Results

### 3.1. Number and Origin of the Samples

A total of 264 vaginal swabs were obtained from the same number of aborted animals in 109 farms throughout the country, including all regions of mainland Greece and the islands of Mytilene, Naxos, and Crete. Among the animal species, 212 were ovine and 52 were caprine. A mean number of 2.4 swabs per farm were collected (2.5 in sheep and 2.2 in goat farms).

### 3.2. Pathogen Detection with the Multiplex qPCR-HRM Technique

#### 3.2.1. Prevalence and Distribution of Identified Pathogens

At least one abortogenic pathogen was detected in 205 of the examined samples (77.7%). Data regarding their prevalence and distribution are presented below.

The results provided in Table 1 clearly indicate a high prevalence of *C. burnetii* and *Chlamydophila* spp. in the studied group, while the prevalence of *C. burnetii* infection was significantly higher in goats than in sheep (Fisher’s exact test, *p* = 0.0004). Moreover, six other abortogenic agents were identified at lower rates. Interestingly, all samples positive for *Brucella* spp. and *T. gondii* were ovine. *L. monocytogenes* and *Leptospira* spp. were not detected.

#### 3.2.2. Polymicrobial Infections

A total of 94 mixed infections were identified: 68 in ovine samples and 26 in caprine. Detailed data are presented in Table 2.

The results in Table 2 highlight the significant prevalence of mixed infections in the examined animals; 35.6% (94/264) of all samples and 45.9% (94/205) of samples in which abortifacient pathogens were identified. Interestingly, at least one of the two pathogens, *C. burnetii* or *Chlamydophila* spp., was implicated in all cases (94/94) of mixed infections. Furthermore, the *C. burnetii* + *Chlamydophila* spp. combination was by far the most common, identified in almost one in two of the mixed infection cases (46.8%, 44/94). The rate of mixed infections was 32.1% (68/212) in sheep and 50.0% (26/52) in goats regarding the total examined samples, while in positive animals the respective rates were 40.9% (68/166) and 66.7% (26/39).

### 3.3. Coxiella burnetii and Chlamydophila spp.

In this subsection, the results of the two methods that assess the same binary outcome are presented without access to a gold-standard method. The results are presented in two-way contingency tables of frequencies with rows and columns indicating the categories of response for each technique (Table 3 and Table 4).

Table 5 presents the values of different indices of test agreement or concordance between measurements. It was concluded that near-perfect agreement between the two methods was achieved (in the range 0.81–0.99, according to [27,33]). Moreover, the application of McNemar’s test to the data implies that there is not enough evidence to show a difference between the two techniques (*C. burnetii* McNemar’s test, *p* = 1.00; *Chlamydophila* spp. McNemar’s test, *p* = 0.227).

Finally, the results related to the problem of testing equivalence or non-inferiority in a matched-pairs design with binary responses and without access to a gold standard are presented.

The 90% confidence intervals for the difference p_1_ − p_2_ based on the four different methods are presented in Table 6. Note that all the CIs are within (−0.05, 0.05) so the methods are equivalent to a predefined δ = 0.05, at significance level a = 0.05.

The results of Nam’s [34] testing procedure are presented in Table 7. According to Liu et al. [30], at significance level a = 0.05, based on the intersection-union principle, the two one-sided tests procedure declares equivalence of the two techniques if the Lower Test Statistic is greater than 1.64 and the Upper Test Statistic is less than −1.64.

Moreover, it follows that the alternative procedure is not inferior to the reference procedure if the Lower Test Statistic is greater than 1.96. Thus, according to these rules, the equivalence of the two methods is confirmed. Moreover, the alternative procedure (HRM-qPCR) is claimed to be not inferior to the reference procedure (qPCR Triplex), since the value of the lower test statistic is greater than 1.96.

In the 19 samples where a disagreement for *C. burnetii* results was observed, the average Ct value of positive HRM-qPCR samples was 36.14 (min 35.38–max 36.98) and the average Ct value of samples positive to qPCR Triplex was 36.69 (min 33.86–max 39.7), demonstrating a low DNA load in these samples. In the 10 samples where a disagreement for *Chlamydophila* spp. results was observed, the average Ct value of samples positive to qPCR Triplex was 36.60 (min 35.09–max 38.81), also demonstrating a low bacterial load in these samples.

### 3.4. Other Pathogens

Table 8 shows the verification results of samples positive for other pathogens from a private diagnostic laboratory. Most positive results were verified, with the exception of *Brucella* spp.; Almost half of the samples (*n* = 7) samples were positive for the HRM-qPCR assay regarding presence of Brucella spp., but were not verified by the relevant commercial qPCR kit. These samples on the HRM-qPCR assay had an average Ct value of 36.27 (min 36.27–max 37.1), indicating a relatively low DNA load.

## 4. Discussion

The accomplished diagnostic approach using the novel HRM-qPCR assay and vaginal swab samples was simple, rapid, and cost-effective, and thus applicable in routine veterinary practice, allowing the collection of sufficient data on the most important pathogens implicated in abortive cases and helping practitioners to apply appropriate therapeutic and preventive measures. A comparison of the novel technique with well-established commercial qPCR assays demonstrated a near-perfect agreement of the results obtained. However, some discrepancies in results were observed, as mentioned previously; the Ct values observed in these samples demonstrate the low microbial load, which in combination with the use of different commercial PCR kits unavoidably impact the consistency of the results and apparently influence the diagnostic sensitivity. This issue can be addressed to some extent with normalizing Ct values across various assays and instruments, ensuring more accurate comparisons and results.

The results of this study demonstrated the high prevalence of both *C. burnetii* and *Chlamydophila* spp. in small ruminant farms throughout Greece. Moreover, the prevalence of *C. fetus* and percentage of mixed infections in the examined group are undoubtedly noteworthy. Finally, to our knowledge, this is the first report of the molecular identification of both *C. fetus* and *N. caninum* from small ruminant abortion cases in the country.

The high detection rates of *C. burnetii* and *Chlamydophila* spp. in sheep and goats with reproductive issues are in accordance with previous reports from southern Europe [2,11,35]. Nonetheless, in other studies, the main pathogen identified was *T. gondii* [36,37]. Furthermore, in serological studies conducted in Greece, comparatively lower positivity was documented for *C. burnetii* (8–17%) [38,39,40]. However, this difference could be a result of the investigated group in this study, since samples were obtained only from individual animals after abortions occurred, and thus, a higher positive rate is expected.

Additionally, the rate of *C. burnetii* detection was significantly higher in caprine samples (Table 1). Similarly, a higher percentage of seropositive goats than sheep was identified by Filioussis et al. in four different regions of Greece (14.4% vs. 8%) [41].

The high prevalence of *C. burnetii* highlights its importance as an abortogenic agent in both sheep and goats. Recently, massive exposure of dairy cattle farms to the pathogen was also reported in Greece (at least one seropositive animal was identified in 25 out of 28 examined farms) [42]. The zoonotic potential of human infection through airborne transmission from animal reservoirs, especially domestic ruminants, has created serious concerns [43]. In approximately two out of three cases where *C. burnetii* or *Chlamydophila* spp. were detected, mixed infections occurred (65.1% and 63.4%, respectively). Therefore, a strong indication is provided that these agents act concurrently in a flock, with other pathogens as well. In each case, the diagnosis of the agents responsible for abortion requires further investigation, such as serological tests, histopathological examination, microbial quantification techniques, and immunohistopathology. Finally, the discrepancy of the results regarding these two pathogens in 19 cases, when comparing HRM-qPCR and qPCR triplex, can be explained by the use of different kits and the low microbial load as depicted by the observed Ct values in these samples, as mentioned previously.

*Campylobacter fetus* was, surprisingly, the third most frequently identified pathogen in this study, since it was detected in 38 cases (33 ovine and five caprine). All caprine cases were mixed infections, whereas *C. fetus* was the only pathogen detected in eight of the ovine cases. *Campylobacter fetus* was only occasionally included in the differential diagnosis of ovine abortions in most previous articles; therefore, data regarding its prevalence in small ruminants are limited. However, in various studies, it has been suggested as a major abortogenic agent for sheep [14] (goats and cattle seem to be less susceptible), whereas it has been obtained from aborted sheep fetuses in variable cases [2,44,45,46]. To our knowledge, this is the first reported molecular identification of *C. fetus* from ovine and caprine abortion cases in the country and possible underdiagnosis needs to be further investigated.

Eleven samples (4.2%) tested positive for *T. gondii*. Our results indicated a low percentage of positive samples in comparison with previous serological studies in Greece [47] and both molecular and serological studies worldwide [48]. Furthermore, according to previous reports, both ovine and caprine toxoplasmosis are endemic in the country [47]. However, it is important to note that placenta and fetal brain are considered the optimal sample type for toxoplasmosis diagnosis, rather than vaginal swabs. Nonetheless, the results of this study demonstrate the potential of vaginal swabs to detect positive cases when recommended samples (fetal brain tissue) are not available. Moreover, all the relevant cases were of ovine origin. This is anticipated because sheep are more susceptible to infection owing to feeding habits and possibly genetic factors [47,49].

Four samples (three ovine and one caprine) were positive for *N. caninum*, and no other pathogens were identified in one of them, even though, similarly to toxoplasmosis, placenta and fetal tissues are the optimal sample type. Although *N. caninum* is not usually included in the differential diagnosis of small ruminant abortions, previous reports suggest that infection with *N. caninum* could affect the reproductive performance of sheep flocks [50,51,52], although it has also been molecularly identified in caprine aborted fetuses [36]. Furthermore, in a seroprevalence study in Greece, specific antibodies against *N. caninum* were detected in 16.8% and 6.9% of examined sheep and goats, respectively [53]. In this regard, the pathogenic potential of *N. caninum* in small ruminants is not negligible and should be further investigated. This is the first published report of the molecular detection of *N. caninum* in abortion samples from small ruminants in Greece. Finally, the identification of *N. caninum* from vaginal swabs is noteworthy and promising because it does not consider the optimal sample type.

*Brucella* spp. was detected in 15 ovine samples; in seven of these samples, the results were not confirmed using a commercially available qPCR kit. However, the low bacterial load, as depicted by the Ct values, may be indicative of the high sensitivity of the HRM-qPCR assay. An analysis of more samples is needed to reach safer conclusions. Detection of the pathogen is of major importance because it constitutes both a virulent abortogenic agent and a serious zoonotic threat. These samples were obtained from 11 different farms, indicating the pathogen distribution. Even though a national program has been running for almost 50 years and an ongoing vaccination and monitoring of animals for the last two decades, *Brucella* spp. has not yet been eradicated [54]. In a recent study, both *B. abortus* and *B. melitensis* were detected in samples from ruminants bred in various regions of the country [55].

*Anaplasma phagocytophilum* was detected in six samples (five ovine and one caprine) and it was the only identified agent in half of them (one caprine and two ovine). This pathogen has been molecularly identified in vectors (*Ixodes ricinus*) obtained from goats in northern Greece [56] and correlated with caprine abortion outbreaks in two farms located in the same region [57]. Therefore, there is a danger of its distribution and infection in both animal and human hosts through tick bites.

Seven cases of *Salmonella* spp. infection were documented. Six of these were samples from sheep, and one was from a goat. In four of the ovine samples, no other agent was identified, indicating *Salmonella* spp. as a possible etiological agent of abortion. This is not surprising since *Salmonella* spp., particularly *Salmonella enterica* subsp. *enterica* serovar Abortusovis, is a potential abortigenic agent, especially in sheep [1]. In an older study conducted in Greece, the pathogen was identified in both ovine and caprine farms at variable rates [58].

No agents were detected in 59 cases (22.3%). This could be associated with infectious causes of other etiology (e.g., abortogenic viruses), or non-infectious causes of abortion, such as toxins, nutritional factors, or climatic conditions. Similar results have been reported in other studies [2].

A significant aspect of the aforementioned results was the rate of identified mixed infections (45.4% of cases with pathogen detection). This result indicates that a considerable percentage of aborting small ruminants were concurrently infected with more than one abortifacient agent. Similarly, several recent studies suggest that ovine and caprine abortions should be partially attributed to mixed infections [11,24,35,37,59,60,61,62]. The rate of mixed infections in this study was significantly higher in goats, corresponding to two of three positive samples (26/39), compared to two of five cases in sheep (68/166). A higher prevalence of caprine mixed infections was also identified by Hazlett et al. [60], whereas in another study from Portugal, the relevant cases were conversely distributed [63]. Nevertheless, interpretation of the rates of concurrent infections presented in different studies is challenging because specific agents and diagnostic assays were included in each investigation; therefore, it is expected that some cases were not identified.

Notably, all 94 cases of mixed infections in this study included *C. burnetii*, *Chlamydophila* spp., or both. These pathogens are prevalent in mixed infections, as reported in previous studies [11,24,35,60,63]. Animals infected with at least one of these pathogens are more likely to be co-infected with more abortogenic agents. It could be assumed that their presence makes animals more vulnerable to secondary infections. This could be associated with their similar lifecycles in ruminant flocks and the pathogenesis procedures when pregnant ewes and does are infected. In particular, although *C. burnetii* and *Chlamydophila* spp. belong to different species, they share many common characteristics, such as their strong tropism to trophoblasts, strategies to evade host cells and avoid immune recognition, the creation and maintenance of a replicative intracellular niche [8,64], their interaction with the host cell, and the pathogenesis of experimental infections [65]. Another possible explanation could be that the high endemicity of these pathogens may be the cause of the high detection prevalence, but only one of these agents or other infectious or non-infectious factors may be the real reason for abortion. More investigation is needed in this aspect.

This study had certain limitations. Initially, only vaginal swabs were examined. However, this is not considered the optimal sampling method for all investigated pathogens, specifically for *T. gondii* and *N. caninum*, hence these pathogens might be underdiagnosed. Molecular identification was the only technique carried out; a complete investigation should include more assays, such as immunological and histopathological tests, cultures, etc. However, a large number of samples originating from a widely distributed sampling region were investigated and tested for major abortogenic pathogens. Therefore, significant epidemiological data were obtained in this study. Finally, for the not commonly detected pathogens, only positive HRM-qPCR results were verified due to study cost limitations; verification of negative results could possibly demonstrate some undiagnosed cases, even though molecular techniques are characterized by very high sensitivity.

In reference to future perspectives, extended sampling and data collection as part of a broader surveillance study is essential to provide detailed epidemiological data. This investigation should include samples from all regions and from more than one reproductive period to decrease the impact of specific regional and climatic factors or seasonal outbreaks. Therefore, verification of possible predisposing factors should be included in the analysis. Quantification of the microbial load, especially in cases of mixed infections, will add important information and depth to the interpretation of infectiousness. Finally, supplementary examination of fetal or placental tissue samples by PCR, histopathology, immunofluorescence, or immunohistochemistry and comparison with vaginal swab results could provide useful information regarding data interpretation in the future.

## 5. Conclusions

The utilization of a novel multiplex HRM-qPCR analysis allows fast and accurate simultaneous detection of the important abortogenic pathogens in small ruminants, which creates new potential to deal with the complexity of ovine and caprine infectious abortions and the requirement for a comprehensive investigation of relevant cases. In this study, we demonstrated that *C. burnetii* and *Chlamydophila* spp. are the prevalent abortogenic agents of small ruminants detected in Greece, and their distribution throughout the country is wide. However, more uncommonly reported pathogens, such as *C. fetus* and *N. caninum*, were identified in certain cases using the novel assay; therefore, their implication in abortions could be underestimated and should be further investigated. Additionally, a significant percentage of infections were polymicrobial. More extended sampling, thorough molecular investigation, and epidemiological data collection will provide sufficient data for circumstantial analysis and safe conclusions on the role of various abortogenic agents in small ruminant production.

## Figures and Tables

**Table 1 microorganisms-12-01675-t001:** Prevalence of abortogenic pathogens in examined samples per animal species. Statistically significant results are shown in bold (*p* < 0.005).

Pathogen	Number of Samples			Fisher’s Exact *p* Value
	Ovine (%)	Caprine (%)	Total (%)	
*A. phagocytophilum*	5 (2.4)	1 (1.9)	6 (2.3)	*p* = 1.0
*Brucella* spp.	15 (7.1)	0 (0)	15 (5.7)	***p* = 0.0475**
*C. fetus*	33 (15.6)	5 (9.6)	38 (14.4)	*p* = 0.3781
*Chlamydophila* spp.	87 (41.0)	25 (48.1)	112 (42.4)	*p* = 0.4339
*C. burnetii*	92 (43.4)	37 (71.2)	129 (48.9)	***p* = 0.0004**
*Leptospira* spp.	0 (0)	0 (0)	0 (0)	-
*L. monocytogenes*	0 (0)	0 (0)	0 (0)	-
*N. caninum*	3 (1.4)	1 (1.9)	4 (1.5)	*p* = 0.5865
*Salmonella* spp.	6 (2.8)	1 (1.9)	7 (2.7)	*p* = 1.0
*T. gondii*	11 (5.2)	0 (0)	11 (4.2)	*p* = 0.1289
Total positive samples	166 (78.3)	39 (75.0)	205 (77.7)	*p* = 0.5830
No detection	46 (21.7)	13 (25.0)	59 (22.3)	

**Table 2 microorganisms-12-01675-t002:** Prevalence and etiologic agents of mixed infections identified in this study.

Etiologic Agents	Number of Cases
*C. burnetii* + *Chlamydophila* spp.	44
*C. burnetii* + *Chlamydophila* spp. + *C. fetus*	8
*C. burnetii* + *Chlamydophila* spp. + *Brucella* spp.	4
*C. burnetii* + *Chlamydophila* spp. + *A. phagocytophilum*	2
*C. burnetii* + *Chlamydophila* spp. + *N. caninum*	2
*C. burnetii* + *Chlamydophila* spp. + *C. fetus* + *Brucella* spp.	1
*C. burnetii* + *C. fetus*	14
*C. burnetii* + *T. gondii*	4
*C. burnetii* + *A. phagocytophilum*	1
*C. burnetii* + *Brucella* spp.	1
*C. burnetii* + *N. caninum*	1
*C. burnetii + Salmonella* spp.	1
*C. burnetii* + *C. fetus* + *Brucella* spp.	1
*Chlamydophila* spp. + *C. fetus*	5
*Chlamydophila* spp. + *Brucella* spp.	3
*Chlamydophila* spp. + *C. fetus* + *Brucella* spp.	1
*Chlamydophila* spp. + *Salmonella* spp. + *Brucella* spp.	1
Total	94

**Table 3 microorganisms-12-01675-t003:** *Coxiella burnetii* HRM-qPCR and qPCR Triplex Contingency table.

	qPCR Triplex Negative (%)	qPCR Triplex Positive (%)	Total
HRM-qPCR Negative	126 (92.6)	9 (7.0)	135 (51.1)
HRM-qPCR Positive	10 (7.4)	119 (93.0)	129 (48.9)
Total	136 (100.0)	128 (100.0)	264 (100.0)

**Table 4 microorganisms-12-01675-t004:** *Chlamydophila* spp. HRM-qPCR and qPCR Triplex Contingency table.

	qPCR Triplex Negative (%)	qPCR Triplex Positive (%)	Total
HRM-qPCR Negative	144 (98.0)	8 (6.8)	152 (57.6)
HRM-qPCR Positive	3 (2.0)	109 (93.2)	112 (42.4)
Total	147 (100.0)	117 (100.0)	264 (100.0)

**Table 5 microorganisms-12-01675-t005:** Indices of test agreement or concordance between measurements.

Index	*C. burnetii*	*Chlamydophila* spp.
Proportion of observed agreement or Rand index	92.8%	95.8%
Positive agreement	92.2%	97.3%
Negative agreement	93.3%	94.7%
Average positive agreement (Dice–Sørensen measure)	92.6%	95.2%
Average negative agreement (Dice–Sørensen measure)	93.0%	96.3%
Cohen’s kappa (k)	0.856 (*p* < 0.001)	0.915 (*p* < 0.001)

**Table 6 microorganisms-12-01675-t006:** The 90% confidence interval for the difference p1 − p2 based on four methods.

Method	*C. burnetii*	*Chlamydophila* spp.
Nam’s score	(−0.0248, 0.0327)	(−0.0431, 0.0019)
Wilson’s score as modified by Newcombe	(−0.0239, 0.0315)	(−0.0403, 0.0025)
Wald Z	(−0.0233, 0.0309)	(−0.0395, 0.0016)
Wald Z with continuity correction	(−0.0272, 0.0347)	(−0.0433, 0.0054)

**Table 7 microorganisms-12-01675-t007:** The results of Nam’s [34] testing procedure.

Nam’s Score	Value	Lower Test Statistic	Lower Probability Level	Upper Test Statistic	Upper Probability Level	TOST Probability Level
*C. burnetii*	0.0038	2.835	0.0023	−2.473	0.0067	0.0067
*Chlamydophila* spp.	−0.0189	2.020	0.0217	−3.981	<0.0001	0.0217

**Table 8 microorganisms-12-01675-t008:** Commercial qPCR results on HRM-qPCR positive samples.

Pathogen	*C. fetus* (%, CI95)	*Brucella* spp. (%, CI95)	*T. gondii* (%, CI95)	*Salmonella* spp. (%, CI95)	*A. phagocytophilum* (%, CI95)	*N. caninum* (%, CI95)
qPCR Positive	35/38 (92.1, 78.6–98.3)	8/15 (53.3, 26.6–78.7)	11/11 (100.0, 71.5–100)	7/7 (100.0, 59.0–100)	4/6 (66.7, 22.3–95.7)	4/4 (100.0, 39.8–100)

## Data Availability

The raw data supporting the conclusions of this article will be made available by the authors on request.

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
