# Peer review of "Molecular Investigation of Small Ruminant Abortions Using a 10-Plex HRM-qPCR Technique: A Novel Approach in Routine Diagnostics"

_microorganisms, 2024, doi:10.3390/microorganisms12081675_

Round 1
Reviewer 1 Report
Comments and Suggestions for Authors
Comments to the Authors
In this study, the authors reported the difference ruminant abortogenic pathogens in sheep and goats throughout Greece. Results indicated a high prevalence of Coxiella burnetii and Chlamydophila spp., which were detected in 48.9% and 42.4% of the vaginal swabs, respectively. Mixed infections occurred in 35.6% of the animals examined. This technique allows for the simultaneous detection of many abortogenic pathogens in an accurate and cost-effective assay. Detection of uncommon or not previously reported pathogens in various cases indicates that their role in ovine and caprine abortions may be underestimated. However, there some major revisions should be clarified or made in the manuscript, as following:
1. Line93,”2.1 sample collection”, authors should describe the growth status and living conditions of samples.
2. Line 217, table I, besides the pathogens listed in Table 1, are there any other microorganisms were detected in the samples?
3. The results provided in Table 1 clearly indicate a high prevalence of C. burnetii and Chlamydophila spp. in the studied group, and experiments about the two pathogens were specially carried out. What is the relationship between the importance and content of pathogens? The pathogens of higher content were more serious harm on the samples? Authors should descript the relationship.
Comments on the Quality of English LanguageModerate editing of English language required
Author Response
In this study, the authors reported the difference ruminant abortogenic pathogens in sheep and goats throughout Greece. Results indicated a high prevalence of Coxiella burnetii and Chlamydophila spp., which were detected in 48.9% and 42.4% of the vaginal swabs, respectively. Mixed infections occurred in 35.6% of the animals examined. This technique allows for the simultaneous detection of many abortogenic pathogens in an accurate and cost-effective assay. Detection of uncommon or not previously reported pathogens in various cases indicates that their role in ovine and caprine abortions may be underestimated. However, there some major revisions should be clarified or made in the manuscript, as following:
- Line 93,”2.1 sample collection”, authors should describe the growth status and living conditions of samples.
Response: We thank the reviewer for the comment. Relevant text has been added, please see lines 96-99.
- Line 217, table I, besides the pathogens listed in Table 1, are there any other microorganisms were detected in the samples?
Response: We thank the reviewer for the comment. Samples were not cultured; they were immediately used for DNA extraction and application of the HRM-qPCR assay. So, the only pathogens that could be detected were the ten pathogens that are included in the assay.
- The results provided in Table 1 clearly indicate a high prevalence of C. burnetii and Chlamydophila spp. in the studied group, and experiments about the two pathogens were specially carried out. What is the relationship between the importance and content of pathogens? The pathogens of higher content were more serious harm on the samples? Authors should descript the relationship.
We thank the reviewer for the comment, please see lines 396-422 where some special biological characteristics of the pathogens have been added that describe the relationship of these pathogens and probably explain their high prevalence. Also a relevant phrase was added about the need to further investgate the load content especially in mixed infections, in future study, performing quantification techniques, please see lines 440-441.
Reviewer 2 Report
Comments and Suggestions for Authors
Gouvias et.al., evaluated the HRM coupled with qPCR technique to detected abortion causing pathogens in sheep and goats. This study is important to discover and develop cost-effective high throughput technology in detecting aborting pathogens with high diagnostic accuracy. Authors can consider few of my few minor comments to improvise the quality.
Minor
1. As rightly mentioned in discussion, this study has many limitations especially comparing the claiming results with gold standard test such as culture. I believe the authors would consider comparing with additional diagnostic assays in the future. I would recommend reconsidering the using the terms “game changing approach” in the title. I felt it is a pure speculative and overstated.
2. Authors followed HRM-qPCR analysis and compared with reference PCR kits. In the discussion section, the authors should address the challenge posed by using different PCR kits, which can impact the consistency of Ct values and apparently influence the test sensitivity. They should also emphasize the importance of normalizing Ct values across various assays and instruments to ensure accurate comparisons and results.
3. While Ct values offer a semi-quantitative estimation of bacterial or viral load, the authors could consider incorporating quantitative load measurements. Exploring the relationship between Ct values, load would add depth to the interpretation of infectiousness.
4. In table 3 and 4, there are a few samples showed disagreement between HRM-qPCR and qPCR triplex. There should be some discussion on this.
5. Maintain the consistency while quoting bacterial genus and species names. Full form at first time (eg: Campylobacter fetus) and followed by abbreviations (C. fetus).
6. Typo in line 284. “presence”
Comments on the Quality of English LanguageMinor editing is required.
Author Response
Gouvias et.al., evaluated the HRM coupled with qPCR technique to detected abortion causing pathogens in sheep and goats. This study is important to discover and develop cost-effective high throughput technology in detecting aborting pathogens with high diagnostic accuracy. Authors can consider few of my few minor comments to improvise the quality.
Minor
- As rightly mentioned in discussion, this study has many limitations especially comparing the claiming results with gold standard test such as culture. I believe the authors would consider comparing with additional diagnostic assays in the future. I would recommend reconsidering the using the terms “game changing approach” in the title. I felt it is a pure speculative and overstated.
Response: We thank the reviewer for the comment, it is our intention to continue the study in future reproductive periods in Greece (starting usually on September/October) and compare/combine results with other than molecular techniques as well, e.g. serology and culture results. We agree with the reviewer on the title, title has changed to Molecular Investigation of Small Ruminant Abortions using a 10-plex HRM-qPCR Technique: A Novel Approach in Routine Diagnostics.
2. Authors followed HRM-qPCR analysis and compared with reference PCR kits. In the discussion section, the authors should address the challenge posed by using different PCR kits, which can impact the consistency of Ct values and apparently influence the test sensitivity. They should also emphasize the importance of normalizing Ct values across various assays and instruments to ensure accurate comparisons and results.
Response: We thank the reviewer for the comment, and we totally agree. A relevant text was added please see lines 297-302.
3. While Ct values offer a semi-quantitative estimation of bacterial or viral load, the authors could consider incorporating quantitative load measurements. Exploring the relationship between Ct values, load would add depth to the interpretation of infectiousness.
Response: We thank the reviewer for this comment and we agree. Relevant reference has been added in lines 440-441.
4. In table 3 and 4, there are a few samples showed disagreement between HRM-qPCR and qPCR triplex. There should be some discussion on this.
Response: We thank the reviewer for this comment and we agree. Relevant reference has been added in lines 330-333.
5. Maintain the consistency while quoting bacterial genus and species names. Full form at first time (eg: Campylobacter fetus) and followed by abbreviations (C. fetus).
Response: We agree with the reviewer, we have made the appropriate changes in the manuscript, leaving the full form of the microbial names only at the first time of reference or when the sentence starts with the microbial name.
6. Typo in line 284. “presence”
Response: Thank you for the comment, error has been corrected.